# Devitalizing Effect of High Hydrostatic Pressure on Human Cells—Influence on Cell Death in Osteoblasts and Chondrocytes

**DOI:** 10.3390/ijms21113836

**Published:** 2020-05-28

**Authors:** Janine Waletzko, Michael Dau, Anika Seyfarth, Armin Springer, Marcus Frank, Rainer Bader, Anika Jonitz-Heincke

**Affiliations:** 1Department of Oral, Maxillofacial and Plastic Surgery, University Medical Center Rostock, 18057 Rostock, Germany; michael.dau@med.uni-rostock.de; 2Department of Orthopedics, Biomechanics and Implant Technology Research Laboratory, University Medical Center Rostock, 18057 Rostock, Germany; anika.seyfarth@med.uni-rostock.de (A.S.); rainer.bader@med.uni-rostock.de (R.B.); anika.jonitz-heincke@med.uni-rostock.de (A.J.-H.); 3Medical Biology and Electron Microscopy Center, University Medical Center Rostock, 18057 Rostock, Germany; armin.springer@med.uni-rostock.de (A.S.); marcus.frank@med.uni-rostock.de (M.F.); 4Department Life, Light & Matter, University of Rostock, 18059 Rostock, Germany

**Keywords:** high hydrostatic pressure, devitalization, decellularization, allografts, regenerative medicine, bone and cartilage regeneration

## Abstract

Chemical and physical processing of allografts is associated with a significant reduction in biomechanics. Therefore, treatment of tissue with high hydrostatic pressure (HHP) offers the possibility to devitalize tissue gently without changing biomechanical properties. To obtain an initial assessment of the effectiveness of HHP treatment, human osteoblasts and chondrocytes were treated with different HHPs (100–150 MPa, 250–300 MPa, 450–500 MPa). Devitalization efficiency was determined by analyzing the metabolic activity via WST-1(water-soluble tetrazolium salt) assay. The type of cell death was detected with an apoptosis/necrosis ELISA (enzyme-linked immune sorbent assay) and flow cytometry. Field emission scanning electron microscopy (FESEM) and transmission electron microscopy (TEM) were carried out to detect the degree of cell destruction. After HHP treatment, the metabolic activities of both cell types decreased, whereas HHP of 250 MPa and higher resulted in metabolic inactivation. Further, the highest HHP range induced mostly necrosis while the lower HHP ranges induced apoptosis and necrosis equally. FESEM and TEM analyses of treated osteoblasts revealed pressure-dependent cell damage. In the present study, it could be proven that a pressure range of 250–300 MPa can be used for cell devitalization. However, in order to treat bone and cartilage tissue gently with HHP, the results of our cell experiments must be verified for tissue samples in future studies.

## 1. Introduction

Musculoskeletal disorders are the world’s leading cause of chronic pain and impaired physical function, which result in loss of life quality [1]. In particular, the treatment of bone-cartilage defects is challenging.

Reasons for a defective or unstable bone structure are on the one hand systemic bone loss due to osteoporosis and on the other hand local trauma or cancerous diseases [2,3]. Often, bone can regenerate itself, but in the case of larger bone defects, where whole segments are missing, this process fails [4]. The gold standard to heal bone defects is still the transplantation of autologous material, which has the advantage that rejections can be avoided [5]. However, the material is naturally limited, and the transplantation of autologous tissue always requires a second surgical treatment which leads to the problem of donor site morbidity [5]. In contrast to autografts, the availability of allografts is less limited. However, the remodeling of allografts into the recipient tissue is critical. Numerous allografts are degraded by recipient’s tissue instead of promoting bone remodeling [6].

Cartilage defects can lead to osteoarthritis when left untreated [7]. One of the main problems of cartilage regeneration is the avascularity of cartilage tissue and low cell turnover, which limit the ability of cartilage healing and self-regeneration [8]. Additionally, for the treatment of cartilage defects, no optimal healing method has been found since regeneration approaches resulted mostly in fibrous tissue instead of hyaline extracellular matrix [9]. The transplantation of scaffolds loaded with mesenchymal stem cells seems to be the most promising approach so far. Nevertheless, an immunological reaction of the recipient in response to the transferred stem cells cannot be completely ruled out, and previously used scaffolds can hardly imitate the complex physiology of osteochondral tissue [8,10]

A promising alternative for patients with musculoskeletal disorders could be the transplantation of allografts which morphologically correspond to the recipient tissue. Additionally, allogenic tissues have only a low potential of immunological response compared to xenogeneic grafts. For the preparation of allografts, strong chemical or physical methods are currently used, which are associated with a significant reduction in biomechanics and a change in the biological behavior regarding remodeling [11]. High hydrostatic pressure (HHP) technology which is widely used in the food industry for the decontamination of food with simultaneous retaining properties such as taste and vitamins, could be a way to devitalize tissues while maintaining their biomechanical properties [12]. If HHP is suitable for providing replacement materials, the kind of cell devitalization should be taken into account during tissue processing. Based on the level of the applied pressure, cells react in either an apoptotic or a necrotic manner [12]. Pressures around 200 MPa seem to induce apoptosis while pressures higher than 300 MPa are associated with necrotic-like pathways [12]. Induction of necrosis provokes a release of proinflammatory molecules and danger signals which can further trigger the immunological response [13]. With a guided initiation of apoptosis using HHP the immunological potential of allografts might be reduced without the need to use other tissue-destructive methods.

The aim of this study was the identification of necessary HHP ranges to initiate apoptosis in human osteoblasts and chondrocytes and establish cell-specific treatment protocols. To assess the potential of HHP treatment for allograft processing, the effects of different HHP ranges on cell survival and cell death were analyzed. Hence, human osteoblasts and chondrocytes were treated with three different HHPs, ranging from 100–150 MPa, 250–300 MPa, and 450–500 MPa, to determine cell viability and HHP-dependent cell death pathways. Additionally, different methodological approaches were carried out to further characterize cell apoptosis and necrosis. To avoid a loss of treated cells due to the experimental setting, the HHP treatment was performed on freshly pelleted cells. Moreover, field emission scanning microscopy (FESEM) and transmission electron microscopy (TEM) were used to provide a detailed insight into cell structures after HHP treatment. Finally, cell specific characterization of HHP treatment served as an indication to transfer cell-specific protocols to tissues in a next step to generate treatment protocols for allogenic tissue such as en bloc bone grafts.

## 2. Results

### 2.1. Cellular Activity

In both cell types, no significant difference in the metabolic activity between the control group and the group treated with 100–150 MPa was detectable (Figure 1). In contrast, HHPs ranging between 250–300 MPa and higher led to a significant decrease in the cell activity in comparison to the control group and the group treated with the lowest HHP range (osteoblasts: *p* = 0.0429 (control vs. 250–300 MPa); *p* = 0.0161 (control vs. 450–500 MPa); *p* = 0.0285 (100–150 MPa vs. 250–300 MPa); *p* = 0.0107 (100–150 MPa vs. 450–500 MPa); chondrocytes: *p* = 0.0155 (control vs. 250–300 MPa); *p* = 0.0150 (control vs. 450–500 MPa)). However, a significant difference in the metabolic activity of cells treated with 250–300 MPa and 450–500 MPa was not detectable.

### 2.2. Induction of Cell Death Following Exposure to HHP

Cell death analysis after HHP treatment was carried out via Cell Death Detection ELISA. In human osteoblasts, HHP exposure at the lowest range (100–150 MPa) led to only a slight increase in necrosis and apoptosis without significant differences compared to untreated cells. In contrast, a significant increase in necrosis (Figure 2a; *p* = 0.0206) and apoptosis (Figure 2b; *p* = 0.0022) of human osteoblasts was shown after HHP treatment of 250–300 MPa. In addition, HHP treatment of 450–500 MPa also resulted in a significant increase in necrosis compared to the control group (*p* = 0.0233) while this HHP range apparently had no effect on osteoblasts regarding apoptosis (Figure 2a,b).

In Figure 2c,d the apoptosis and necrosis rates of human chondrocytes after HHP treatment are shown. A significant increase in necrosis took place in response to an HHP range of 250–300 MPa compared to untreated cells (*p* = 0.0184). In addition, the results of apoptosis (Figure 2d) revealed that the lowest HHP (100–150 MPa) induced mostly apoptosis while higher HHP ranges had no apoptotic effect. However, these data were not statistically significant.

### 2.3. Differentiation of Necrosis, Apoptosis, and Late Apoptosis Following HHP Treatment of Pelleted Cells

Since Cell Death ELISA data revealed that HHP treatment of 250–300 MPa had the strongest effect on the tested cell types, a more detailed cell death analysis took place via flow cytometry. In comparison to this pressure, pelleted cells were also treated with a pressure of 100–150 MPa again for analysis via flow cytometry. The used gating strategy is shown in Figure 3 on the basis of untreated human osteoblasts. After gating the cells from the cell debris with the side scatter height (SSC-H) and forward scatter height (FSC-H) channels, fluorescence (FL)4-H on the y-axis for apoptosis and FL2-H on the x-axis for necrosis were set. Then, four quadrants (Q)1 to Q4 were adjusted, which represent annexin single positive cells (Q1; apoptosis), annexin/propidium iodide (PI) double positive cells (Q2; late apoptotic and necrotic cells), PI single positive cells (Q3; necrosis), and annexin/PI double negative cells (Q4; vital cells), respectively.

The majority of untreated groups of both cell types could be detected as double negative (alive, around 80%). The remaining parts of the cells were distributed between apoptotic, necrotic and late apoptotic/necrotic detection for both osteoblasts and chondrocytes. A similar distribution could be observed for cells treated with 100–150 MPa for 10 min (Figure 4a,b). Human osteoblasts (Figure 4a) were significantly late apoptotic/necrotic following HHP treatment of 250–300 MPa in comparison to the control group (*p* < 0.0001), while the live, apoptotic and necrotic cells were less than 5%. In contrast to this, human chondrocytes were detected as necrotic and late apoptotic/necrotic in a wide range after treatment with 250–300 MPa (Figure 4b). However, a significant increase in late apoptotic/necrotic cells after treatment with 250–300 MPa in comparison to the control group could be detected (*p* < 0.0001), and an increase in necrotic cells after HHP treatment of 250–300 MPa in comparison to the control group (*p* = 0.0041) could be observed.

### 2.4. Evaluation of the Cellular Damage of Osteoblasts after HHP Treatment Based on Field Emission Electron Microscopy (FESEM) and Transmission Electron Microscopy (TEM)

Imaging techniques such as FESEM and TEM were used to visualize the HHP effects on cell structures. Therefore, osteoblasts were treated with HHP (100–150 MPa and 250–300 MPa for 10 min each) and subsequently prepared for FESEM and TEM analysis.

Untreated human osteoblasts could be detected via FESEM with an uneven cell surface with distinct elevations which appeared to be smooth (Figure 5a). In the sections, analyzed by TEM, cellular structures such as the nucleus, mitochondria, and the endoplasmic reticulum were observed (Figure 5b). Between the control group and the group treated with 100–150 MPa, only slight differences could be detected. Again, the cell surface appeared to be intact in FESEM, and TEM analyses revealed distinct structures as described before (Figure 5c,d). Only after treatment of human osteoblasts with 250–300 MPa, the cell surface appeared to be altered and became a sponge-like structure covered with small blebs (Figure 5e). While in the other two groups, the intracellular structures were clearly visible with TEM, these structures were destroyed following HHP treatment with 250–300 MPa (Figure 5f). Based on a comparison of the cell diameters of the differently treated cells, the untreated and 100–150 MPa treated groups had a diameter of approximately 20 μm while most of the cells treated with 250–300 MPa seemed to shrink approximately by one fifth.

## 3. Discussion

The need for tissue replacement materials, e.g., in the head and neck area, has increased in the last years, particularly with regard to the increasing amount of cancerous diseases [14]. The gold standard of using autologous material has the disadvantage of being limited, and patients have to undergo two operations. Allografts, which are also common, have to be treated with strong chemicals to reduce the immunogenic potential. At the same time, the matrix integrity is lost [15].

Synthetic materials, which are not limited in availability, often cannot withstand the biomechanical strain and are resorbed quickly, which makes them unsuitable for use in major bone defects [5]. The treatment of allogenic material with high hydrostatic pressure (HHP) could be a gentle and effective alternative to already existing methods for providing decellularized tissue replacement materials without negative effects on the tissue matrix [12]. The aim of this study was to evaluate an optimal HHP protocol for different cell types (osteoblasts and chondrocytes) regarding devitalization and induction of apoptosis or necrosis. Rivalain et al. already described the effect of different HHPs on mammalian cells and observed, that after treatment of around 200 MPa, cells underwent apoptosis whereas a treatment higher than 300 MPa resulted in necrosis [12].

We were able to characterize the influence of high hydrostatic pressure on cell survival and cell death in human osteoblasts and chondrocytes. We further aimed to distinguish differing HHP-effects when cells were exposed as cell pellets. The first observation was that the lowest HHP range of 100–150 MPa had no significant effect on the metabolic activity of all tested cell types. In contrast, treatment of cells with HHP ranging from either 250–300 MPa or 450–500 MPa had the same effect regarding the significant reduction of metabolic activity. This observation was a first hint that HHP treatment of 250 MPa and higher destroyed cells and resulted in devitalization. The previous work of Hiemer et al. [8] showed that an HHP of 480 MPa led to successful devitalization of hyaline cartilage cells. However, the limitation of the work was that this devitalizing effect with respect to apoptosis and necrosis was not investigated. As mentioned before, this point is of high relevance because cell death is decisive for the induction of immunological processes. The requirement for tissue replacement materials is low to negligible immunological potential for the acceptor. Therefore, the devitalization process should not have a necrotic effect because this would lead to inflammatory responses regarding the release of, e.g., damage-associated molecular patterns (DAMPS) which further stimulate the pro-inflammatory response [13]. To analyze the type of cell death induced by HHP, necrosis and apoptosis were first detected by ELISA. The collected data showed that the untreated control groups also underwent apoptosis and necrosis partly, which can be attributed to the fact that the controls were exposed to non-standard conditions (e.g., room temperature) during this time that the treated groups underwent the HHP treatment. Aside from this finding, HHD treatment of 100–150 MPa had no significant effect in comparison to the control group. A balanced apoptosis- and necrosis-inducing effect on osteoblasts was observed at 250–300 MPa, whereas an HHP treatment of 450–500 MPa induced primarily necrosis. In contrast, an HHP range of 100–150 MPa induced apoptosis in chondrocytes derived from hyaline cartilage whereby necrosis was already induced by a range from 250–300 MPa. The most effective HHP treatment should first and foremost lead to metabolically inactive cells that have, ideally, not yet undergone necrosis. Thus, taking the results of both, the WST-1 assay and ELISA-derived apoptosis and necrosis analysis into consideration, these findings suggest that an HHP of 200–300 MPa may be the most effective for gentle and noninflammatory cell devitalization. Analysis via flow cytometry, providing a more precise separation of vital, necrotic, apoptotic, and late apoptotic/necrotic cells, revealed that most osteoblasts were vital after HHP treatment of 100–150 MPa, while exposure to 250–300 MPa led to necrosis and apoptosis only in a small number of cells. Moreover, most of these cells were detected as being in a late apoptotic/necrotic condition. Identical observations were made for chondrocytes. We assumed that cells treated with the higher HHP range were apoptotic at an earlier time point and underwent necrosis. This result can be explained, among other things, by the fact that there was a time delay between the HHP treatment and the subsequent FACS analysis due to the availability of equipment at different locations. Therefore, it cannot be excluded that necrosis was induced in the cells after a short time. Additionally, the absence of any phagocytic cells in the experimental setting led to the absence of cell debris removal. Consequently, the membrane of the treated cells became permeable over time, and intracellular contents could be released, which might further stimulate the immunological response [13]. Additionally, permeable and destroyed cell membranes as well as cellular components were found in TEM analysis of osteoblasts treated with 250–300 MPa. In summary, the results support the idea that in the early phase after HHP, apoptosis was induced following a pressure of 250–300 MPa, but the initiation of intracellular material packaging was incomplete or aborted. As confirmed by FACS analyses, the cells were in a late apoptotic/necrotic state, which led to an uncontrolled destruction of cell components in the intracellular cavity, which was additionally supported by TEM. Interestingly, these results were not observed for the lower HHP range of 150–200 MPa. Therefore, we assumed that either a reduction of the HHP level or a shortening of the HHP exposure time could initiate apoptotic devitalization. This aspect has to be confirmed in future studies.

The aim of this study was to examine the cell-typical reactions to various HHP treatments to assess the tissue-specific reactions to HHP in the next step. Furthermore, our results on devitalization efficiency and cell death detection will help to better evaluate the results of a previous animal study on devitalized autografts [10]. Applied pressure of 480 MPa to osteochondral cylinders led to an extensive tissue remodeling process following tissue re-transplantation. However, excellent integration and revitalization were detected [10]. Possible reasons for these findings might be the autograft transplantation in the experimental setting and the animal-specific immune system, which plays an important role in tissue integration. For better understanding, it is even more important to perform targeted analysis of HHP-mediated cell death which is of high relevance for the subsequent translation of specific tissue types into the clinical use of allografts. It is also important to study additional tissue-specific cells to prove whether the respective high pressure successfully devitalizes all resident cells in the tissue through apoptosis. Taking late apoptosis/necrosis into account, rapid purification of the HHP-processed tissue is still mandatory so that cell residues can be removed gently and in sufficient time. These aspects will be analyzed in more detail in further studies.

## 4. Materials and Methods

### 4.1. Cell Culture

Human osteoblasts from femoral heads (*n* = 5) and human chondrocytes from hyaline knee cartilage (*n* = 5) were isolated from patients undergoing total joint replacements. Prior to isolation of human osteoblasts and chondrocytes, informed consent and ethical approval (A2010-10A (osteoblasts) and A2009-17 (chondrocytes) were obtained from the ethics committee of the University of Rostock, Germany) including IRB information. Harvesting and isolation of cells took place after patients signed agreement forms following the previously described protocols [16,17,18]. We have got the written informed consent from all patients. The isolated osteoblasts were cultivated in a 25 cm^2^ culture flask with a volume of 8 mL osteogenic medium (Dulbecco’s modified Eagle’s medium (DMEM), PAN-Biotech, Aidenbach, Germany) supplemented with 10% fetal calf serum (FCS, PAN-Biotech), 1% amphotericin B, 1% penicillin/streptomycin, 1% HEPES buffer, 50 μg/mL ascorbic acid, 10 mM β-glycerophosphate, and 100 nM dexamethasone (all: Sigma–Aldrich, Munich, Germany). Human chondrocytes were cultivated in 25 cm^2^ culture flasks containing 8 mL DMEM medium (Gibco^®^ Invitrogen, Paisly, UK) containing 10% FCS, 1% amphotericin B, and 1% penicillin/streptomycin. In addition, chondrocytes were supplemented with 50 µg/mL ascorbic acid. Incubation took place in a humidified atmosphere of 5% CO_2_ and at 37 °C. The respective medium was changed every second day so non-adherent cells could be aspirated. At a confluence of 100%, which was checked by light microscopy, the cells were transferred in a 75 cm^2^ culture flask, and further culture took place under the same conditions as described above. Cells in the third passage were used for the experiments.

### 4.2. High Hydrostatic Pressure Treatment

To treat cells with high hydrostatic pressure (HHP), the medium was discarded and adherent cells were detached with 2 mL trypsin per 75 cm^2^ flask. After an incubation time of 3 min at 5% CO_2_ and 37 °C, the reaction was stopped with 6 mL DMEM. The cell suspension was collected and centrifuged at 118× *g* for 8 min. The medium was discarded, and the obtained cell sediment was resuspended in 1 mL DMEM. After cell counting, 50,000 cells were transferred as duplicates into 2 mL CryoTubes (ThermoFisher Scientific, Waltham, MA, USA), filled up with the corresponding medium, and centrifuged at 118× *g* for 8 min to generate freshly pelleted cells. These cell pellets were treated with different HHPs ranging from 100–150 MPa, 250–300 MPa, and 450–500 MPa for 10 min using an HHP device (HDR-100, RECORD GmbH, Koenigsee, Germany). After HHP treatment, the tubes were centrifuged again at 118× *g* for 8 min; 200 µL of the supernatants were discarded, and the cells were incubated for 24 h under standard cell conditions. HHP-untreated cells served as controls; these cells were exposed to the same test conditions.

### 4.3. Metabolic Activity Analysis

After an incubation time of 24 h, HHP-treated and untreated cells were centrifuged at 118× *g* for 8 min. The supernatants were discarded, and the metabolic activity cells was analyzed by incubating the cells for 1.5 h under standard culture conditions with 250 μL of a 1:10 dilution of WST-1 reagent (Takara, Bio, Saint-Germain-en-Laye, France) and the respective cell culture medium. The amount of formed formazan dye, which correlates directly with the number of metabolic active cells, was determined by an extinction measurement using a Tecan-Reader Infinite^®^ 200 Pro (Tecan, Maennedorf, Switzerland); (absorption: 450 nm, reference: 600 nm).

### 4.4. Analysis of Cell Death

For cell death detection, supernatants of exposed and unexposed cells were collected and stored at −20 °C for further necrosis detection by Cell Death Detection ELISA (Roche, Penzberg, Germany). For apoptosis detection, the residual cells were incubated with 200 μL lysis buffer provided by the kit for 30 min at room temperature. The cell lysates were centrifuged at 200× *g* for 10 min, and the supernatants without the cell debris were stored at −20 °C for further use.

To analyze the kind of cell death, all supernatants were thawed and transferred as duplicates into a streptavidin-coated microtiter plate with the related positive, negative, and background controls. In each well, 80 μL immunoreagent, containing incubation buffer, anti-histone-biotin, and anti-DNA-POD, all made available by the kit, were added to each well. The microtiter plate was covered with the cover foil and incubated for 2 h at room temperature on the shaker at 300 rpm. The supernatants were then discarded, and the plate was washed three times with 250 μL incubation buffer. Afterwards, the buffer was aspirated thoroughly, and 100 μL ABTS Solution was added. After an incubation period of 10 min at room temperature on the shaker at 250 rpm, a color reaction was visible and it was stopped by adding 100 μL ABTS Stop Solution. The quantification took place in a Tecan-Reader Infinite^®^ 200 Pro at a wavelength of 405 nm and a reference wave length of 490 nm.

For precise analysis of cell death after HHP treatment, human osteoblasts and chondrocytes were analyzed by flow cytometry with the APC Annexin-V Apoptotis Detection Kit with propidium iodide (PI) (Biolegend, San Diego, CA, USA). Cells were cultured as described above (all cell types *n* = 5) and a cell number of 2 × 10^5^ cells were treated with an HHP of 100–150 MPa and 250–300 MPa for 10 min, whereas cell pellets were treated and analyzed. Afterwards, cells were centrifuged at 118× *g* for 8 min. The supernatants were discarded and the cells were washed with 1 mL autoMACS^®^ Running Buffer (Myltenyi Biotec, Bergisch Gladbach, Germany) at 1400 rpm for five minutes. A mastermix of 5 μL Annexin V FITC and 10 μL PI per sample was added, and the cells were incubated for 15 min at room temperature in darkness. Afterwards, 100 μL Annexin Binding buffer was added, and analysis took place with the FACS Calibur (BD, Franklin Lakes, NJ, USA). For the evaluation of the results, FloJo (BD, Franklin Lakes, NJ, USA) was used, and the percentage of positive cells was assessed.

### 4.5. Analysis of Cellular Damage by Field Emission Electron Microscopy (FESEM) and Transmission Electron Microscopy (TEM)

Human osteoblasts and human chondrocytes (100,000 cells each) were transferred into 2 mL CryoTubes (ThermoFisher), filled up with the corresponding medium, and centrifuged at 118× *g* for 8 min. These cell pellets were treated with different pressure ranges (w/o HHP, 100–150 MPa and 250–300 MPa). After treatment, tubes were again centrifuged at 118× *g* for eight minutes and the supernatant was discarded. The cells where then fixed with fixation buffer (1% paraformaldehyde, 2.5% glutaraldehyde, 0.1 M sodium phosphate buffer, pH 7.3) and stored at 4 °C. Afterwards, cells were washed with sodium phosphate buffer, adhered to glass coverslips previously coated with poly-L-lysine (Co. Sigma) for one hour and then dehydrated with an acetone series, followed by critical point drying using CO_2_ (Emitech K850/Quorum Technologies Ltd., East Sussex, UK). The glass coverslips were mounted on aluminum sample holders and coated with gold under vacuum (SCD 004, Baltec, Balzers, Liechtenstein). With a field emission scanning electron microscope (MERLIN VP Compact, Carl Zeiss, Oberkochen Germany), images were taken from the selected regions (applied detector: HE-SE2; accelerating voltage: 5.0 kV; working distance: 5.2 mm).

For TEM preparation, fixed specimens were washed in 0.1 M sodium phosphate buffer (pH 7.3), and cell pellets were enclosed in 3% low melting agarose (Fluka, Munich, Germany) in water at 40 °C by centrifugation in Eppendorf tubes. After staining with 1% osmiumtetroxide (Roth GmbH, Karlsruhe, Germany) for 2 h, the specimens were dehydrated through an ascending series of acetone prior to embedding in Epon resin (Serva, Heidelberg, Germany). Resin infiltration started with a 1:1 mixture of acetone and resin overnight, followed by pure resin for 4 h. After transfer to rubber casting moulds, specimens were cured in an oven at 60 °C for at least 48 h. Resin blocks were trimmed using a Leica EM Trim 2 (Leica Microsystems, Wetzlar, Germany). Ultrathin sections (approx. 70–90 nm) were cut with a Leica UC7 ultramicrotome using a diamond knife (Diatome, Nidau, Switzerland). Ultrathin sections were mounted on formvar-coated copper grids and were contrasted with uranyl acetate and lead citrate. Ultrastructure was inspected with a Zeiss EM902 electron microscope operated at 80 kV (Carl Zeiss, Oberkochen, Germany). Digital images were acquired with a side-mounted 1x2k FT-CCD Camera (Proscan, Scheuring, Germany) using iTem Camera control imaging software (Olympus Soft Imaging Solutions, Muenster, Germany).

### 4.6. Statistics

All results are expressed as medians with interquartile ranges (25–75%) and whiskers. As this is the first study on this topic, no a priori power calculation could be performed. Testing for statistical significance was done by one- or two-way ANOVA. *p*-values ≤ 0.05 were seen as significant. All statistical analyses were performed with GraphPad Prism Version 7 (GraphPad Software, San Diego, CA, USA).

## Figures and Tables

**Figure 1 ijms-21-03836-f001:**
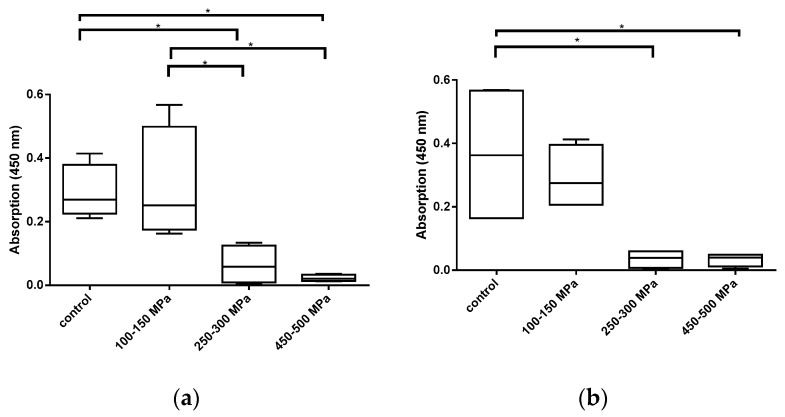
Metabolic activity of (**a**) human osteoblasts (*n* = 4) and (**b**) human chondrocytes (*n* = 4) following treatment with different HHPs ranging from 100–150 MPa, 250–300 MPa, and 450–500 MPa. After HHP exposure, cells were incubated at 37 °C and 5% CO_2_ over a period of 24 h. Subsequently, metabolic activity was determined with water-soluble tetrazolium salt (WST-) 1 assay. Data are presented as boxplots. Significant differences between stimulation groups were determined via one-way ANOVA: * *p* ≤ 0.05.

**Figure 2 ijms-21-03836-f002:**
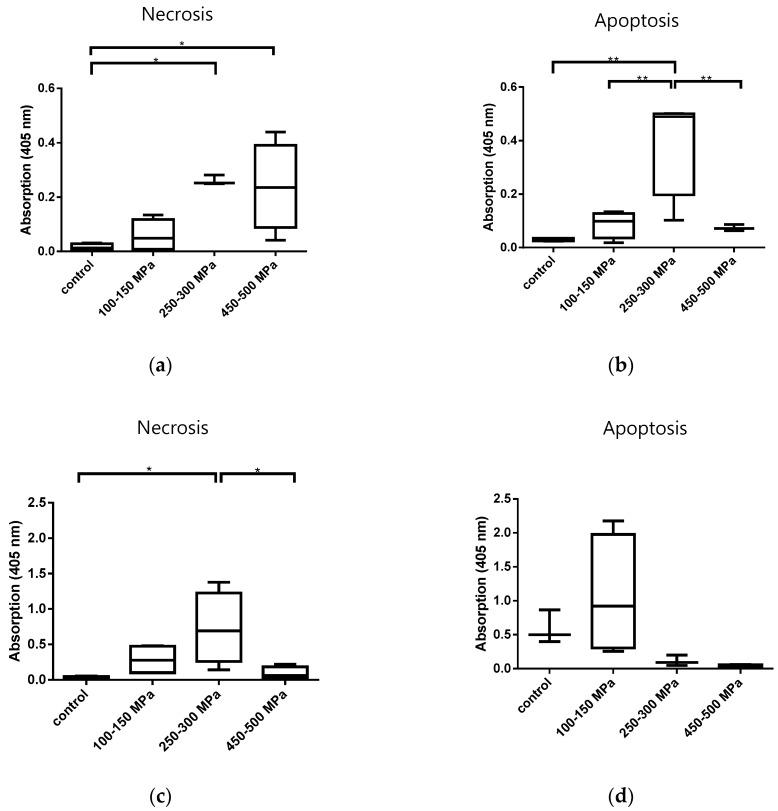
Analysis of cell death induced by HHP treatment of (**a**,**b**) human osteoblasts (*n* = 4), and (**c**,**d**) human chondrocytes (*n* = 4). Graphics (**a**,**c**) show necrosis and graphics (**b**,**d**) show apoptosis, both detected via Cell Death Detection ELISA. After HHP treatment, the supernatants were collected for necrosis detection. The residual cells were lysed for 30 min with 200 µL lysis buffer provided by the kit. After centrifugation at 118× *g* for 8 min, supernatants were collected for apoptosis detection. All data are presented as boxplots. Significant differences between groups were determined via one-way ANOVA. * *p* ≤ 0.05; ** *p* ≤ 0.01.

**Figure 3 ijms-21-03836-f003:**
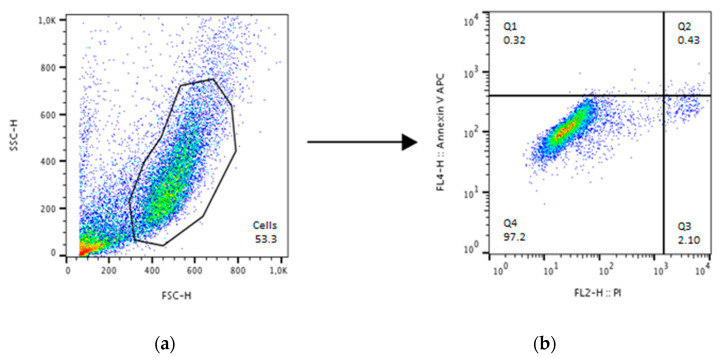
Gating strategy for cell death detection with annexin and propidium iodide (PI) exemplified using untreated human osteoblasts. Analysis took place with FloJo. (**a**) Gating of cells from the debris using the side scatter height (SSC-H) and forward scatter height (FSC-H) channel. This gate was used to detect the different stainings in (**b**) where quadrant (Q)1 represents annexin single positive cells, Q2 annexin/PI double positive cells, Q3 PI single positive cells and Q4 annexin/PI double negative cells.

**Figure 4 ijms-21-03836-f004:**
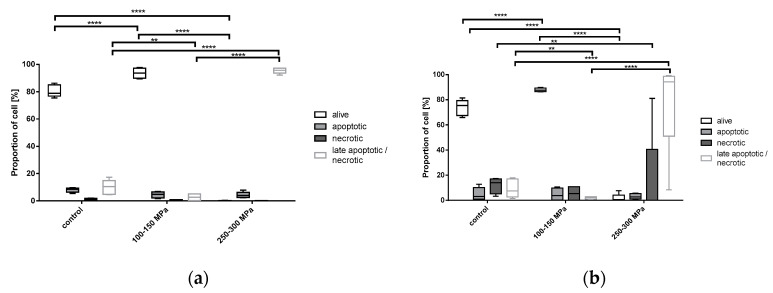
Analysis of cell death detection after HHP treatment (100–150 MPa and 250–300 MPa, 10 min) of (**a**) human osteoblast cell pellets (*n* ≥ 4) and (**b**) human chondrocyte cell pellets (*n* ≥ 5) with annexin and propidium iodide (PI). For each cell type, cell pellets were prepared for HHP treatment. Afterwards, treated cells were washed with autoMACS^®^ Running Buffer, stained with annexin and PI, and analyzed with the FACS Calibur. Data are shown as box plots. Significance between groups was detected via two-way ANOVA. ** *p* ≤ 0,01; **** *p* < 0.0001.

**Figure 5 ijms-21-03836-f005:**
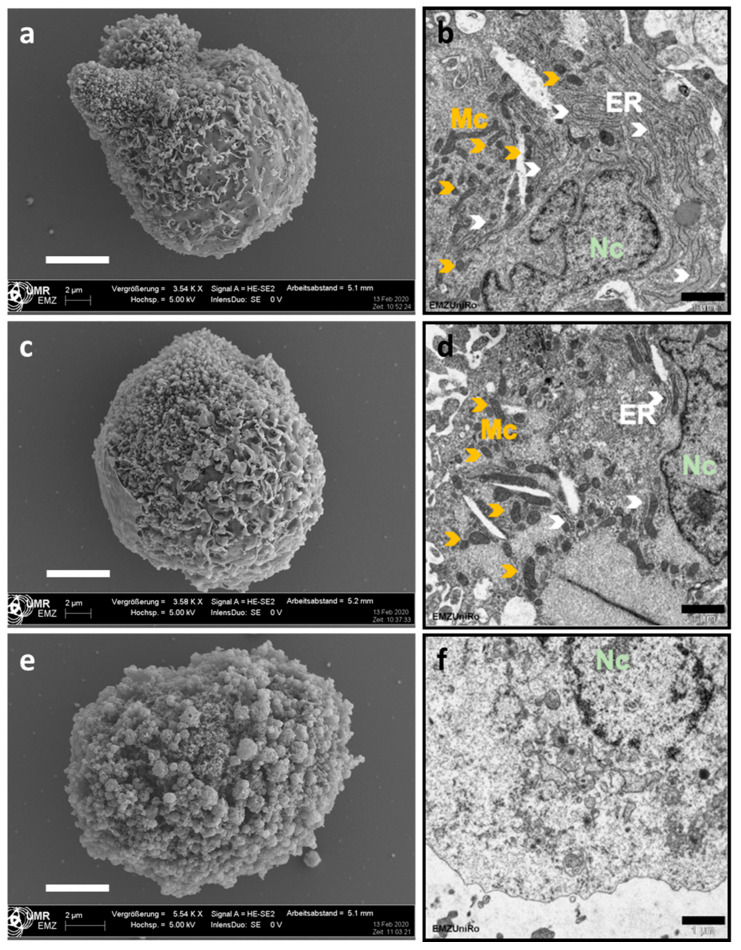
Cell structure analysis of human osteoblasts following HHP treatment. Treated cells (**c**,**d** 150–200 MPa; **e**,**f**: 250–300 MPa) and untreated cells (**a**,**b**) were fixed to carry out field emission electron (**a**,**c**,**e** scale bar: 5 μm) and transmission electron (**b**,**d**,**f**; scale bar: 1 μm) microscopy. Abbreviations: Nc = nucleus (light green), Mc = mitochondrion (orange), ER: endoplasmic reticulum (white).

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
