# Peer review of "Devitalizing Effect of High Hydrostatic Pressure on Human Cells—Influence on Cell Death in Osteoblasts and Chondrocytes"

_ijms, 2020, doi:10.3390/ijms21113836_

Round 1

Reviewer 1 Report

The manuscript aims at analyzing the effect of high hydrostatic pressure on osteoblasts and chondrocytes for bone allografts devitalization. The paper is clear, structured and well-written. However, some points should be clarified.

On the manuscript authors stated the effect of HHP was evaluated on spheroids. However, according to the protocol they described on the materials and methods section they do not treat spheroids with HHP what they really used was just freshly pelleted cells. Could authors clarify this point?

Authors included in the discussion section  “Line 223: HHP of 200-300 MPa is the most effective one for gentle and non-inflammatory cell devitalization” while I guess they mean 250-300 MPa. Further than that, at this pressure there is high necrosis for chondrocytes with no apparent apoptosis according to the ELISAs. Could authors explain why they selected this pressure as the adequate one for both cell types? Moreover, osteoblasts show some metabolic activity at this pressure, could this be a significant problem for the application of HHP in allograft devitalization? Have authors thought on testing an HHP range between 300 and 450?

Authors used the different levels of apoptosis and necrosis after HHP treatment. However, to be able to contextualize the overall effect of HHP over both parameters it would be great if authors could include relevant positive controls able to set up the maximum levels for both parameters.

Authors should more clearly state the relevance of the obtained results for the intended application. In other words, how the experimental settings selected (pelleted cells) can resemble those of allograft tissues characterized by cells surrounded by extracellular matrix? Have authors considered to test the HHP on cells embedded on mature spheroids (with extracellular matrix) instead of just cell pellets?

Figure 2 do not include “c” and “d” on the image. This should be fixed.

Reviewer 2 Report

Dear authors,

Why did you select cells in the third passage for the experiments? Did you characterize osteoblasts and chondrocytes by using immunohistochemical or histological (alizarin red, safranine 0…) controls?

Results

(c) and (d) are missing in Figure 2.

Figure 5 : Why did you only show effect of HHP on osteoblasts by SEM and TEM and not on chondrocytes ?

Please correct reference 2 : Shrivats AR

You must delete the mention of Supplementary Materials in the manuscript.

Best regards
